# Genome-wide association study identifies five risk loci for pernicious anemia

Triin Laisk [1 ✉], Maarja Lepamets[1,2], Mariann Koel[1,2], Erik Abner [1], Estonian Biobank Research Team* & Reedik Mägi[1]

Pernicious anemia is a rare condition characterized by vitamin B12 deficiency anemia due to lack of intrinsic factor, often caused by autoimmune gastritis. Patients with pernicious anemia have a higher incidence of other autoimmune disorders, such as type 1 diabetes, vitiligo, and autoimmune thyroid issues. Therefore, the disease has a clear autoimmune basis, although the genetic susceptibility factors have thus far remained poorly studied. We conduct a genome-wide association study meta-analysis in 2166 cases and 659,516 European controls from population-based biobanks and identify genome-wide significant signals in or near the *PTPN22* (rs6679677, $p = 1.91 \times 10^{-24}$, OR = 1.63), *PNPT1* (rs12616502, $p = 3.14 \times 10^{-8}$, OR = 1.70), *HLA-DQB1* (rs28414666, $p = 1.40 \times 10^{-16}$, OR = 1.38), *IL2RA* (rs2476491, $p = 1.90 \times 10^{-8}$, OR = 1.22) and *AIRE* (rs74203920, $p = 2.33 \times 10^{-9}$, OR = 1.83) genes, thus providing robust associations between pernicious anemia and genetic risk factors.

[1] Estonian Genome Centre, Institute of Genomics, University of Tartu, Tartu, Estonia. [2] Institute of Molecular and Cell Biology, University of Tartu, Tartu, Estonia. A list of authors and their affiliations appears at the end of the paper. ✉email: triin.laisk@ut.ee

B12 deficiency anemia due to intrinsic factor deficiency, also known as pernicious anemia, is characterized by impaired B12 uptake caused by lack of intrinsic factor, a glycoprotein produced by epithelial cells of the stomach lining. Intrinsic factor normally binds B12 and facilitates absorption in the intestinal tract. Pernicious anemia is often caused by autoimmune damage to the stomach lining (autoimmune gastritis) in which case the gastric epithelial lining is damaged or destroyed[1]. The prevalence of pernicious anemia is around 0.1% in populations of European ancestry; however, it is more common in older people (~2% in >60 year olds), and believed to be less prevalent in Asian populations[2]. Symptoms of pernicious anemia range from fatigue to megaloblastic anemia and neurological abnormalities (peripheral numbness, paresthesia, and ataxia) in more serious cases[3].

Pernicious anemia is a complex disease with familial clustering, with a clear autoimmune basis and higher incidence of other autoimmune diseases, such as autoimmune thyroid conditions[4], vitiligo[5], and type 1 diabetes[6] in both patients with pernicious anemia and their relatives[7]. Previous studies have identified genetic variants affecting vitamin B12 levels in the general population[8] or associated with gastric parietal cell autoantibody positivity in type 1 diabetes patients[9], but it is not known whether these associate with pernicious anemia as well. Although studies focusing on HLA serotypes have been conducted for pernicious anemia, the results have been conflicting[7] and currently there is no clear consensus on the HLA alleles or other genetic risk factors predisposing to pernicious anemia.

Here, we conduct a genome-wide association study (GWAS) of vitamin B12 deficiency due to lack of intrinsic factor to evaluate the contribution of genetic variation to the etiology of this disease in a combined dataset of 2166 cases and 659,516 controls from three large population based biobanks—Estonian Biobank (EstBB)[10], UK Biobank (UKBB)[11], and FinnGen study. Our analyses identify five genome-wide significant signals, thus providing evidence for robust associations between pernicious anemia and genetic risk factors.

## Results

In the EstBB, individuals with pernicious anemia were identified using the ICD-10 code D51.0, resulting in 378 cases and 138,207 controls for analysis (prevalence 0.3%). Association testing was carried out with SAIGE 0.38 software which is suitable for phenotypes with a pronounced case-control imbalance[12], adjusting for sex, year of birth, and ten PCs. Individual level data analysis in the EstBB was carried out under ethical approval 1.1-12/624 from the Estonian Committee on Bioethics and Human Research (Estonian Ministry of Social Affairs) and data release N05 from the EstBB. GWAS summary statistics for the UKBB analysis including White British participants were downloaded from the UKBB PheWeb (http://pheweb.sph.umich.edu/SAIGE-UKB). Similarly, cases in the UKBB had been identified using the ICD-10 code D51.0 (754 cases, 390,026 controls; prevalence 0.2%) and SAIGE had been used for association testing, adjusting for sex,

birth year, and the first four PCs. Summary statistics for FinnGen study were obtained from the publicly available R3 release PheWeb (https://www.finngen.fi/en/access_results). In the FinnGen study, we used the endpoint "Vitamin B12 deficiency anemia", which included all the subcodes in the ICD10 D51 diagnosis group (1034 cases, 131,283 controls; prevalence 0.8%). Similarly to other cohorts, SAIGE had been used for association testing, adjusting for sex, age, ten PCs, and genotyping batch. For meta-analysis, we used fixed-effects meta-analysis implemented in GWAMA[13]. Additional details available in Methods section.

**GWAS and candidate gene mapping.** In our GWAS meta-analysis we identified five genome-wide significant ($p < 5 \times 10^{-8}$) associations (Table 1, Fig. 1, and Supplementary Fig. 1) on 1p13.2 (lead signal rs6679677, $p = 1.91 \times 10^{-24}$), 2p16.1 (rs12616502, $p = 3.14 \times 10^{-8}$), 6p21.32 (rs28414666, $p = 1.40 \times 10^{-16}$), 10p15.1 (rs2476491, $p = 1.90 \times 10^{-8}$), and 21q22.3 (rs74203920, $p = 2.33 \times 10^{-9}$), with similar effect estimates (effect heterogeneity was measured using Cochran's test, $p$ values ranging from 0.05 to 0.83) in all analysed cohorts (Fig. 1c and Supplementary Table 1), except for the lead variant on chromosome 2, which is absent in the Finnish data.

The credible sets of most likely causal SNPs at each associated locus were determined using the standard approximate Bayesian finemapping approach[14,15]. When selecting the most likely candidate gene in each associated region, we considered the following criteria—(a) whether the credible set includes a coding variant in any of the nearby genes, (b) whether the signal colocalises with a variant that affects gene expression in COLOC[16] analysis (Supplementary Data 1), and (c) relevant biological functions of the neighboring genes and the mouse phenotypes of corresponding gene knock-outs.

The lead SNP on chr1, rs6679677, is in high LD ($r^2 = 0.96$) with a non-synonymous variant in exon 12 of the PTPN22 gene, a well-described genetic risk variant for autoimmune diseases (rs2476601, $p = 2.82 \times 10^{-24}$) (these are also the only two variants in the credible set; Supplementary Fig. 2). PTPN22 is a known immune regulator gene and this particular variant (rs2476601 A allele) has been associated with an increased risk of several autoimmune diseases, including rheumatoid arthritis, systemic lupus erythematosus, vitiligo, autoimmune thyroid conditions, type 1 diabetes, and others (Supplementary Data 2, 3 and Supplementary Fig. 3). The A allele also increases the risk of pernicious anemia (OR 1.62; 95%CI 1.48–1.78).

On chromosome 2, the credible set included 17 variants. Colocalization analysis showed pernicious anemia GWAS association colocalises with PNPT1 eQTL signal in thyroid tissue in GTEx v8 dataset (posterior probability for shared causal variant PP4 = 0.87; Fig. 2 and Supplementary Data 1), PNPT1 exon expression QTL in monocytes (PP4 = 0.93–0.96), RP11-554J4.1 eQTL in multiple tissues (PP4 = 0.84–0.92), and CCDC104 exon expression QTL in fat and blood (PP4 = 0.81–0.91). Credible set analysis showed that two variants (rs7586115 and rs13420929, $r^2 = 0.7$ with lead signal rs12616502) overlap with PNPT1

**Table 1 Summary of GWAS meta-analysis results for pernicious anemia.**

| Lead variant | chr:pos[a] | EA/ NEA | EAF | $p$ value | OR (95% CI) | Cochran's test $p$ values |
|---|---|---|---|---|---|---|
| rs6679677 | 1:114303808 | A/C | EstBB = 0.140 UKBB = 0.1 FinnGen = 0.147 | $1.91 \times 10^{-24}$ | 1.63 (1.48-1.79) | 0.71 |
| rs12616502 | 2:55809015 | A/G | EstBB = 0.055 UKBB = 0.051 FinnGen = NA | $3.14 \times 10^{-8}$ | 1.70 (1.41-2.05) | 0.84 |
| rs28414666 | 6:32626451 | G/A | EstBB = 0.767 UKBB = 0.790 FinnGen = 0.684 | $1.40 \times 10^{-16}$ | 1.38 (1.28-1.49) | 0.83 |
| rs2476491 | 10:6095410 | A/T | EstBB = 0.688 UKBB = 0.700 FinnGen = 0.755 | $1.90 \times 10^{-8}$ | 1.22 (1.14-1.30) | 0.21 |
| rs74203920 | 21:45714294 | T/C | EstBB = 0.018 UKBB = 0.015 FinnGen = 0.038 | $2.33 \times 10^{-9}$ | 1.83 (1.5-2.29) | 0.05 |

[a]Positions according to GRCh37.

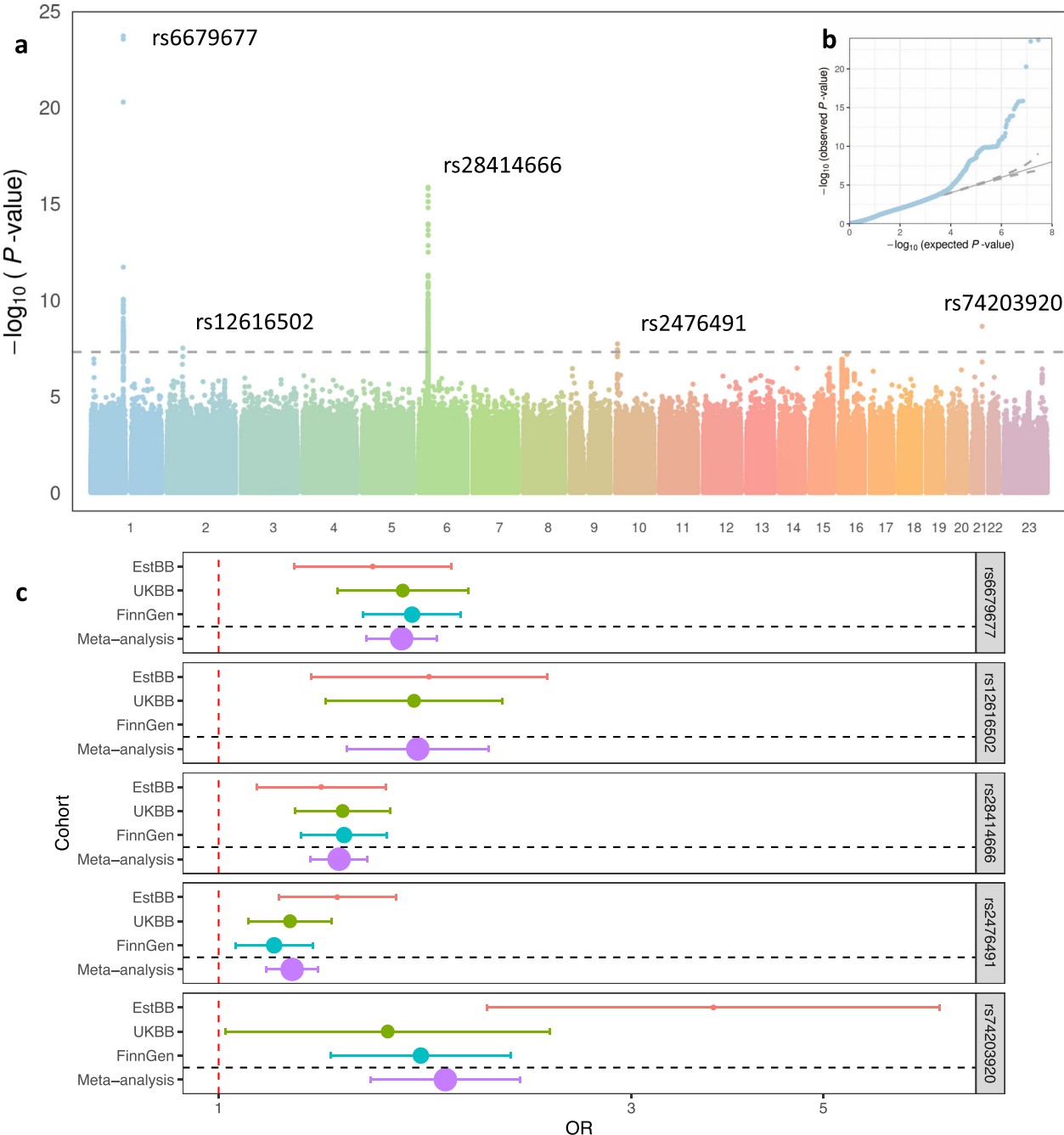

**Fig. 1 Results of the pernicious anemia GWAS meta-analysis. a** Manhattan and **b** QQ plot. *P* values from inverse variance fixed effect meta-analysis, gray dashed line represents the genome-wide significance threshold to account for multiple testing ($p < 5 \times 10^{-8}$); **c** Forest plot of effect estimates for lead variants associated with pernicious anemia. Data were presented as odds ratios and 95% confidence intervals (error bars) for all included cohorts and meta-analysis ($N_{meta} = 661,682$, $N_{EstBB} = 138,585$, $N_{UKBB} = 390,780$, and $N_{FinnGen} = 132,317$). The size of the dot is proportional to the effective sample size, (calculated as 4/ ((1/$N$_cases)+(1/$N$_controls))). Pernicious anemia is defined as ICD10 code D51.0 in EstBB and UKBB and as D51 (vitamin B12 deficiency anemia) in FinnGen.

transcription start site/flanking region or enhancer marks in several cell types and tissues, including T-cells subtypes (Fig. 2). Unlike the lead signal, these two variants are also present in the FinnGen dataset, although statistically not significant (in FinnGen data, rs7586115 $p = 0.51$, OR 1.07 (0.87–1.33); meta-analysis heterogeneity *p* value 0.01). Data from mouse knockouts shows that *Pnpt1tm1a(KOMP)Wtsi/Pnpt1+* male mice exhibit higher mean corpuscular volume (MCV)[17] (Supplementary Data 4 and Supplementary Fig. 4) compared to females, unlike their background strain C57BL/6 in which MCV is similar in males and

females[18]. Increased red blood cell MCV is a common feature in macrocytic anemias, both megaloblastic (caused by B12 deficiency and pernicious anemia) and nonmegaloblastic (caused by diseases such as myelodysplastic syndrome and hypothyroidism)[19]. When we stratified our analysis according to sex (in EstBB and UKBB where we had access to individual level data), we saw a significant difference in effect sizes for the lead signal rs12616502 (OR 2.22 (1.63–3.00) in men, OR 1.39(1.15–1.67) in women, GWAMA gender heterogeneity *p* value 0.01; Supplementary Table 1). No other lead signal exhibited sexual dimorphism (Supplementary

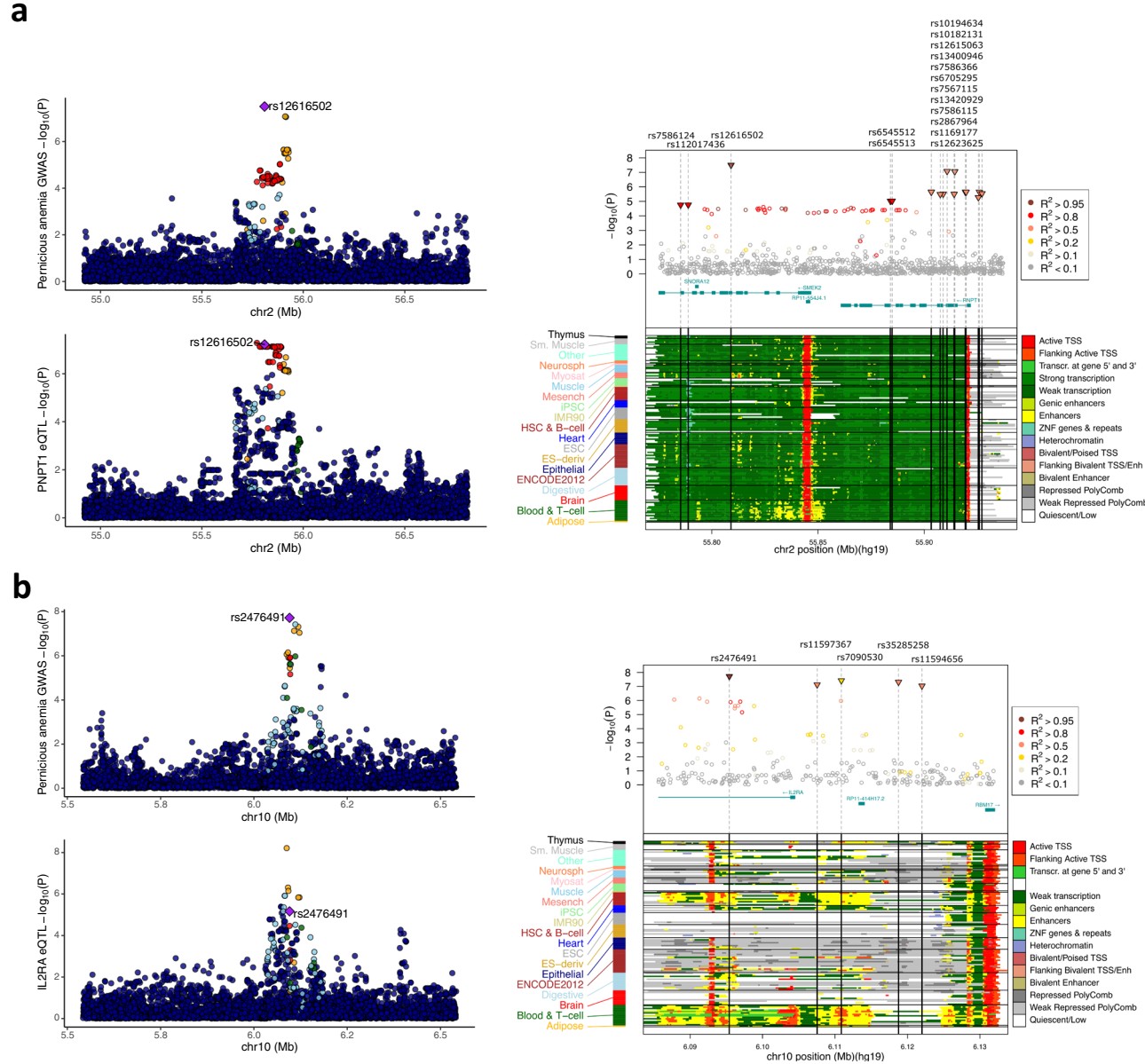

**Fig. 2 Colocalisation analyses and annotation of credible set variants.** Colocalisation of GWAS signals and functional annotation of the credible set variants on chromosomes 2 (**a**) and 10 (**b**). For both panels, the left side depicts colocalisation results (top—GWAS-meta-analysis; bottom—QTL analysis). For all plots, LD is colored with respect to the GWAS lead signal (labeled). The right side shows functional annotation of the credible set variants (top— regional association statistics from meta-analysis. Credible set variants are marked with triangles and the color coding of each variant corresponds to the LD pattern in the region (see legend in the figure); bottom—15-state chromatin marks for 127 samples from Roadmap Epigenomics project; color coding explained in the legend on the figure). **a** GWAS signature for pernicious anemia signal on chromosome 2 co-localizes with the eQTL signal for *PNPT1* (posterior probability for shared causal variant PP4 = 0.87 in thyroid tissue in GTEx v8 dataset). **b** GWAS signature for pernicious anemia signal on chromosome 10 co-localizes with the transcript QTL signal for *IL2RA* (left; posterior probability for shared causal variant PP4 = 0.85 in monocytes).

Table 1). This region has been previously associated with both vitiligo, hypothyroidism, and myelodysplastic syndrome (Supplementary Data 2, 3, and Supplementary Fig. 3).

The third signal, on chromosome 6, is located in the HLA region, a common hub for autoimmune condition associations, downstream *HLA-DQB1*. To further clarify the association signal in the *HLA* region, we used data on a total of 249 imputed *HLA* alleles at four-digit level available for 378 cases and 138,189 controls in the EstBB. Three tested alleles passed the Bonferroni corrected threshold of association (0.05/249 = $2 \times 10^{-4}$), including *HLA-DQB1*06:02 ($p = 1.0 \times 10^{-5}$, OR 1.62 (1.31-2.00)), *HLA-DQA1*01:02 ($p = 3.6 \times 10^{-5}$, OR 1.50 (1.24-1.81), and *HLA-DRB1*15:01 ($p = 3.6 \times 10^{-5}$,

OR 1.57 (1.27-1.94)). After conditioning on these three alleles, we still observed some residual signal in the locus (rs4148874, $p = 3.4 \times 10^{-5}$); however, when we conditioned on the *HLA* region lead signal (rs4148874) in the EstBB, the association with the aforementioned *HLA* alleles lost its significance. The *DRB1*15:01-DQB1*06:02-DQA1*01:02* combination forms the HLA-DR15 haplotype, which is a reported risk factor for multiple sclerosis[20]. HLA-DR15 belongs under the HLA-DR2 group, which has been associated with pernicious anemia in a 1981 study[21]. Unfortunately, imputed *HLA* allele data were not available for other included studies.

On chromosome 10, the sentinel variant rs2476491 is intronic to *IL2RA*. The credible set included an additional four SNPs (Fig. 2).

We found the GWAS signal colocalises with transcription event QTL signal in monocytes (PP4 = 0.84), corresponding to *IL2RA*-205 processed transcript (transcript ID ENST00000644262.1). Of the credible set variants, rs2476491 and rs7090530 overlap with enhancer marks and active TSS flanking region in several cell types, including T-cell subtypes. *IL2RA* encodes the interleukin-2 receptor alpha chain, thus being involved in regulating regulatory T-cells and immune tolerance, as regulatory T cells suppress autoreactive T-cells. Accordingly, this locus has previously been associated with multiple sclerosis, juvenile idiopathic arthritis, vitiligo, and hypothyroidism (Supplementary Data 2).

Finally, the association on chromosome 21, rs74203920, is a missense variant in the *AIRE* gene, a known autoimmune regulator. Mutations in *AIRE* are a known cause of autoimmune polyendocrinopathy syndrome type 1 (APS-1), which is a rare autosomal recessive syndrome, that sometimes includes pernicious anemia among other components[22]. It has also been shown that some *AIRE* mutations located in the SAND and PHD1 domains have a dominant negative effect on wild type AIRE, leading to common forms of autoimmune diseases (incl. pernicious anemia)[22]. rs74203920 leads to Arg471Cys substitution in the PHD2 domain, which is needed for AIRE interaction with proteins involved in chromatin structure/binding or transcription[23]. On a molecular level, the amino acid change could lead to a change in binding partners or affects binding of the stabilizing Zn2+ molecule[23]. Further studies are needed to explore the molecular effects of this amino acid alteration.

To clarify the overlap between genetic regulation of pernicious anemia and natural B12 levels, we did a look-up of eleven variants associated with vitamin B12 levels in a large Icelandic whole genome sequence dataset combined with Danish exome sequencing data (Supplementary Data 5). None of these variants were genome-wide significant in our data, but 8/11 showed nominally significant association *p* values, with the allele decreasing serum B12 levels consistently also increasing the risk of pernicious anemia (Supplementary Data 5). Finally, parietal cell antibody positivity (a biomarker for autoimmune gastritis and pernicious anemia) has been associated with the *ABO* locus at 9q34[24] and T1D risk loci[9]. The lowest *p* value (*p* = 0.005) for the *ABO* region in our meta-analysis was for rs8176760. Six of the nine T1D risk loci showing significant PCA associations were at least nominally significant in our meta-analysis (Supplementary Data 5), including rs2476601 (*PTPN22*), which is one of our top associated variants.

**Associated phenotypes**. We used the individual level data in EstBB to evaluate the association between pernicious anemia and other diseases (defined by ICD-10 codes). According to our analysis, individuals with pernicious anemia have more diagnoses of other anemias and vitamin deficiencies (Fig. 3 and Supplementary Data 6), thyroid problems (thyroiditis and hypothyroidism), gastrointestinal tract diagnoses (gastritis, malignant neoplasm of stomach, intestinal malabsorption, irritable bowel syndrome), but also of vitiligo, dermatitis, osteoporosis, and spontaneous abortion. Majority of these diagnoses reflect the etiology of pernicious anemia (gastritis) or symptoms (skin problems, syncope and collapse, depression, and stomatitis), known comorbidities (vitiligo and thyroid issues[4,5]), or diseases where pernicious anemia is a known risk factor (such as osteoporosis[25] and stomach cancer[26]). The association with spontaneous abortion is interesting, as although there is some evidence B12 deficiency and pernicious anemia could cause recurrent miscarriage[27–29], the data is scarce and the link with spontaneous miscarriage has not been explored in depth.

We also tested the prevalence of other autoimmune diseases among the cases compared to controls. In EstBB, 55.8% of all pernicious cases have at least one other autoimmune diagnosis (23.9% in controls). For comparison, we did a similar look-up for other common autoimmune diseases as well— type 1 diabetes (35.9% vs 22.8%), vitiligo (37.9% vs 23.5%), rheumatoid arthritis (39.3% vs 19.2%), and Hashimoto's thyroiditis (33.7% vs 18.6%). This confirmed the higher incidence of autoimmune diseases in pernicious anemia as well as other autoimmune diseases. A similar look-up in the UKBB data showed that 35% of pernicious anemia cases and 9% of controls had at least one other autoimmune disease from our tested list (Methods).

To check whether the pernicious anemia associations were driven by concomitant autoimmune diseases (mostly vitiligo and thyroid problems, which were also highlighted as significant in the associated diagnoses analysis), we did a look-up for vitiligo, hypothyroidism, and autoimmune thyroid disease associations in our meta-analysis summary statistics (Supplementary Data 5). Since autoimmune diseases can share pathogenic mechanisms, we focused on loci that according to current knowledge do not regulate autoimmune response. For vitiligo, we chose variants annotated to genes associated with melanocyte biology and for thyroid issues we chose the *TSHR* (thyroid stimulating hormone receptor) and *TPO* (thyroid peroxidase) loci and others (Supplementary Table 5)[30]. None of these variants reached a nominal significance in our pernicious anemia GWAS meta-analysis, confirming that the observed associations are not mainly driven by concomitant autoimmune disease.

**Phenotypic effects of rs74203920 (*AIRE*)**. To evaluate the phenotypic effect of the *AIRE* missense variant rs74203920, we conducted a pheWAS analysis for the alternative allele carrier status. In the EstBB dataset, 4882 individuals were either heterozygous or homozygous for the alternative allele. The only diagnosis group showing significantly increased prevalence in the alternative allele carriers was D51 for vitamin B12 deficiency anemias (including D51.0 for pernicious anemia) ($p = 8.6 \times 10^{-6}$, OR = 1.5(1.3–1.7)).

**Discussion**

In the current study we identify five susceptibility loci for pernicious anemia. Candidate gene mapping involving variant annotation, finemapping, and colocalisation analyses, and data from mouse models highlights the involvement of genes with a known role in autoimmune disease (*PTPN22*, *HLA*, *IL2RA*, and *AIRE*).

While for some of the associated regions, the signal could be mapped to missense coding variants with a functional impact on genes with a known role in autoimmune regulation (such as the *PTPN22* and *AIRE* loci), for others we used a combination of finemapping, colocalisation analysis and comparison with data from mouse knockouts to propose the most likely causal variants and genes. This led to the conclusion than *PNPT1* and *IL2RA* are the most likely candidate genes on chromosome 2 and 10, respectively. While *IL2RA* has an established role in immune regulation (it encodes the interleukin-2 receptor alpha chain and regulates Treg cells and immune tolerance), less is known about *PNPT1*. It encodes a polyribonucleotide nucleotidyltransferase that predominately localizes in the mitochondrial intermembrane space and participates in importing RNA to mitochondria. Disrupted PNPT1 function causes accumulation of mitochondrial RNA in the cytoplasm, which leads to immune activation[31]. In most severe forms, this presents as a group of disorders known as type I interferonopathies, which are commonly characterized by autoinflammation and autoimmunity[32]. Additionally, PNPT1 is part of a cascade regulating TET2[33], an important factor related to hematopoiesis and innate immunity[34]. Dysregulation of TET2 can lead to both hematological malignancies as well as autoimmune conditions[33,34]. The effect of nonpathogenic variants that are common in the population and may affect *PNPT1* expression

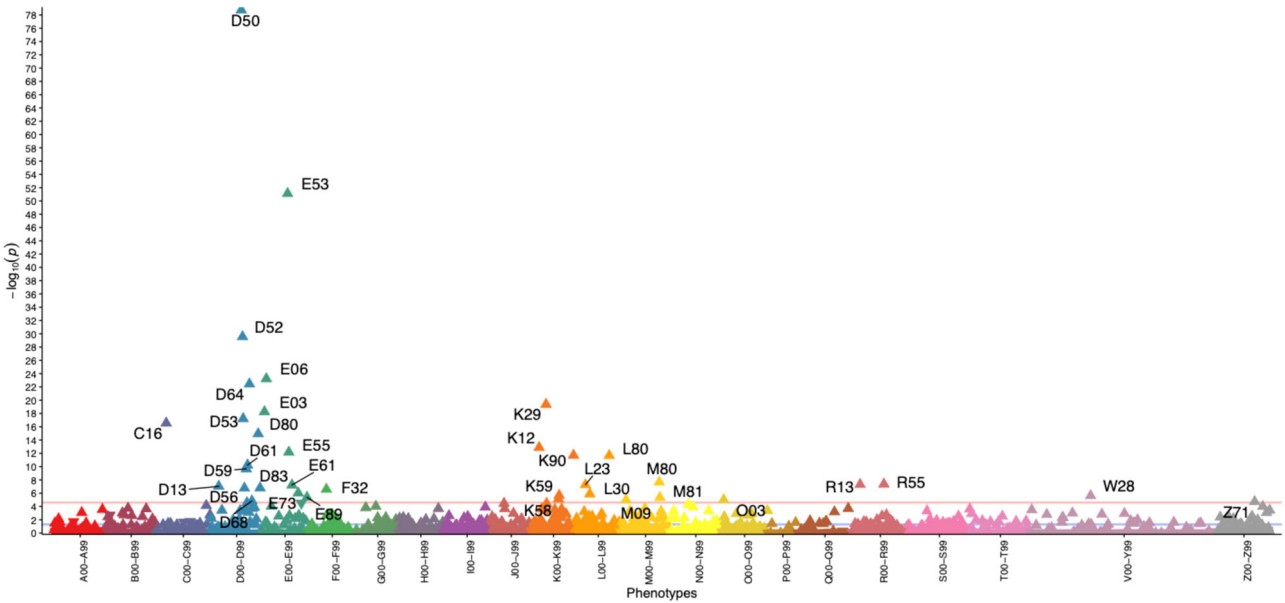

**Fig. 3 Association plot for ICD10 codes associated with pernicious anemia diagnosis in Estonian Biobank.** Each triangle in the plot represents one ICD10 main code and the direction of the triangle represents direction of effect—upward-pointing triangles show increased prevalence of diagnosis code in pernicious anemia cases. Red line— Bonferroni-corrected significance level ($2.5 \times 10^{-5}$).

levels has not been studied in detail. Our study provides evidence for a sexually dimorphic effect of this locus on risk of pernicious anemia, as we observed significantly larger effect sizes in men. In a recent GWAS of blood cell traits, our lead signal for this locus (rs12616502) was nominally associated with MCV ($p = 7.8 \times 10^{-3}$) in European ancestry analysis[35]. We believe the association at this locus together with the association we saw with the *AIRE* missense variant rs74203920 provide new leads for functional studies, as the phenotypic effect of rs74203920 both on molecular and organism level has also not been described before extensively, apart from a recent study which confirms its link with autoimmune disease[36].

In our analysis, cases were identified from population-based biobank data using the ICD-10 code for pernicious anemia (D51.0 or D51 in the FinnGen data). This approach was selected to simplify data analysis, but we acknowledge this approach may have some shortcomings. First, the used summary statistics for FinnGen cohort are for a broader vitamin B12 deficiency phenotype definition compared to other two cohorts (so it likely includes other cases of B12 deficiency). Second, the use of the code may vary in different healthcare systems, increasing heterogeneity of the phenotype. However, we see no significant differences in effect estimates for the reported lead variants, and the identified loci and candidate genes have a clear role in autoimmune regulation, suggesting the vast majority of cases in this study have autoimmune B12 deficiency (pernicious anemia). The potential misclassification of our control subjects as not having pernicious anemia can increase the heterogeneity in the analysed data and attenuate the results towards the null, meaning that either larger datasets with the current phenotype definition or further refinement of the phenotype definition is needed to increase the number of identified loci. At the same time, the prevalence of pernicious anemia in our studied datasets ranged from 0.2–0.8%, which is roughly in line with the expected prevalence of pernicious anemia (0.1% in the general population and >2% in over 60 year olds)[2].

Pernicious anemia is often accompanied by other autoimmune diseases, such as autoimmune thyroid conditions[4], vitiligo[5], and type 1 diabetes[6]. We see similar results in the EstBB dataset, as individuals with pernicious anemia had vitiligo and hypothyroidism significantly more often, and overall,

pernicious anemia cases had other autoimmune diagnoses more often than controls both in EstBB and UKBB. To rule out the confounding effect of concomitant diagnoses (especially vitiligo and hypothyroidism which were significantly more common in cases), we did a look-up for vitiligo, hypothyroidism, and autoimmune thyroid disease associated variants in our meta-analysis summary statistics and saw that loci specific to vitiligo and thyroid issues (melanocyte biology and thyroid function, respectively) were statistically not significant in our analyses, indicating that these diagnoses did not confound our results. In parallel, it is known that autoimmune diseases share genetic risk loci[37], therefore it is not surprising the loci associated with pernicious anemia are pleiotropic and also affect the risk of other autoimmune diseases.

Look-up of genetic variants associated with vitamin B12 levels in our data showed that in general, the allele which decreases serum B12 levels increases the risk of pernicious anemia, although none of these variants was genome-wide significant in our analysis. It cannot be ruled out that, in addition to the autoimmune component, pernicious anemia has some of its origins in the biological regulation of vitamin B12 levels, or alternatively—people with naturally lower B12 levels are diagnosed earlier. This could also mean our dataset of supposedly pernicious anemia cases also includes individuals with other causes of B12 deficiency, or alternatively—the datasets used in the Grarup et al. study[8] of B12 levels in the general population included cases of pernicious anemia. However, as pointed out before, the loci identified in this study and mapped candidate genes have a clear role in autoimmune regulation, suggesting most our analysed cases have autoimmune B12 deficiency (pernicious anemia).

A similar look-up of variants associated with parietal cell autoantibodies (a biomarker of autoimmune gastritis and pernicious anemia) in T1D patients showed that many of these loci are nominally associated with our analysed phenotype as well. The associated loci have a central role in immune regulation (incl *PTPN22*, *CTLA4*, *IFIH1*, *HLA* region, and *SH2B3*). Notably, we did not see an association with the parietal cell autoantibody-associated *INS* locus[9], which again suggests that the genome-wide significant loci we report are not confounded by accompanying

autoimmune disease (such as T1D where *INS* plays a central role in the etiopathogenesis).

This GWAS meta-analysis focused on individuals with pernicious anemia identified from population-based biobanks, thus more detailed analysis on the effect these genetic risk factors have on disease severity or subphenotypes is unfortunately not possible. The analyses included participants of European ancestry, however, analyses in other populations are warranted, since the prevalence of pernicious anemia differs depending on racial background, being more common in people with African or European ancestry, especially from Scandinavia and UK[2]. The analysis of associated phenotypes could potentially provide clinically useful insights into pernicious anemia disease trajectories and offer information for patient management; however, at the moment it only includes participants of the EstBB, with limited follow-up time, therefore further studies are needed.

In summary, our analysis of 2166 cases and 659,516 controls identifies robust risk loci for pernicious anemia in or near candidate genes with a known role in autoimmune conditions (*PTPN22*, *HLA*, *IL2RA*, and *AIRE*) and suggests *PNPT1* as a potential causal gene with possible sexually dimorphic effects in the 2p16.1 locus that needs further validation. The associations between the identified loci and other autoimmune conditions, such as type 1 diabetes, vitiligo, and autoimmune thyroid conditions help to clarify the link between pernicious anemia and its common comorbidities. Analysis of associated diagnoses confirm the association between pernicious anemia and thyroid issues, vitiligo, gastritis, stomach cancer, osteoporosis, and other diagnoses, but also between pernicious anemia and spontaneous abortion.

## Methods
### Cohorts

*Estonian Biobank.* The EstBB is a population-based biobank with over 200,000 participants. The 150 K data freeze was used for the analyses described in this paper. All biobank participants have signed a broad informed consent form. Individuals with pernicious anemia were identified using the ICD-10 code D51.0, and all biobank participants who did not have this diagnosis were considered as controls. Information on ICD codes is obtained via regular linking with the national Health Insurance Fund and other relevant databases[10]. Individuals with pernicious anemia were identified using the ICD-10 code D51.0, resulting in 378 cases (22% males (age at baseline 63.5 ± 15.7 years) and 78% females (54.9 ± 15.9 years)) and 138,207 controls (34% males (43.1 ± 16.2 years), and 66 % females (44.0 ± 16.0 years)) for analysis.

All EstBB participants have been genotyped at the Core Genotyping Lab of the Institute of Genomics, University of Tartu, using Illumina Global Screening Array v1.0 and v2.0. Samples were genotyped and PLINK format files were created using Illumina GenomeStudio v2.0.4. Individuals were excluded from the analysis if their call-rate was <95% or if sex defined based on heterozygosity of X chromosome did not match sex in phenotype data. Before imputation, variants were filtered by call-rate <95%, HWE *p* value < 1e-4 (autosomal variants only), and minor allele frequency <1%. Variant positions were updated to b37 and all variants were changed to be from TOP strand using GSAMD-24v1-0_20011747_A1-b37.strand.RefAlt.zip files from https://www.well.ox.ac.uk/~wrayner/strand/ webpage. Prephasing was done using Eagle v2.3 software[38] (number of conditioning haplotypes Eagle2 uses when phasing each sample was set to:–Kpbwt=20000) and imputation was done using Beagle v.28Sep18.793[39] with effective population size ne = 20,000. Population specific imputation reference of 2297 WGS samples was used[40].

Association analysis was carried out using SAIGE (v0.38)[12] software implementing mixed logistic regression model with LOCO option, using sex, year of birth, and ten PCs as covariates in step I.

*UK Biobank.* The UKBB is a prospective cohort of 502,637 individuals aged 37–73 recruited in 2006–2010 from across the UK, who completed detailed questionnaires regarding sociodemographic and lifestyle characteristics and their medical history and had a clinical assessment. Additional information about medical conditions (both existing at baseline and occurring during follow-up) has been obtained through linking with hospital admission and mortality data. Full details of the study have been reported in Sudlow et al.[11]. Publicly available GWAS summary statistics downloaded from the UKBB PheWeb [http://pheweb.sph.umich.edu/SAIGE-UKB] were used for the analysis. Briefly, the PheWeb includes GWAS summary statistics for ICD code-based traits extracted from electronic health records. Phenotypes have been classified into 1,403 broad PheWAS codes, including pernicious anemia (PheCode 281.11), defined using the ICD-10 code

D51.0 and excluding other anemias under the PheCodes 280–285.99. Genetic analyses have been carried out using SAIGE[12]. All variants in the UKBB summary statistics file had an imputation INFO score ≥0.3.

For additional analyses requiring individual-level data (cohort descriptive statistics, sex-stratified analysis of lead signals, and look-up of additional autoimmune diseases in pernicious anemia cases), we used data under the application 17085. No additional ethics approval were needed for this dataset and UKBB's Ethics and Governance Council provides guidelines for conducting studies with this dataset. We focused on samples of genetically confirmed British European ancestry. We excluded individuals who had withdrawn their consent, were labeled with poor heterozygosity or missingness as defined by UKBB, had excess (>10) relatives, were not included in autosome phasing, had putative sex chromosome aneuploidy, or had sex mismatch between self-reported and genotype data. Pernicious anemia cases were extracted using the ICD10 D51.0 code in HES (Hospital Episodes and Spells) data (downloaded on July 11, 2020). Since the phenotype data was extracted at different timepoints and using slightly different (exclusion) criteria, the follow-up analyses included a larger number of pernicious anemia cases (*n* = 1192) compared to the original GWAS analysis (*n* = 754). The descriptive characteristics of the follow-up dataset in the UKBB were: cases—384 men (age 61.9 ± 15.7 years) and 808 women (age 59.1 ± 7.7 years), controls 187,056 men (age 57.1 ± 8.1 years) and 219,756 women (age 56.7 ± 7.9 years).

*FinnGen.* FinnGen is a public–private partnership project combining data from Finnish biobanks and electronic health records from different registries. After a 1-year embargo, the FinnGen summary stats are available for download. In this study, we used the results from the FinnGen release R3, which includes data from 135,638 individuals and more than 1800 disease endpoints. FinnGen individuals have been genotyped with Illumina and Affymetrix arrays and imputed to the population-specific SISu v3 importation reference panel. Genetic association testing has been carried out with SAIGE[12]. The FinnGen disease endpoint "Vitamin B12 deficiency anemia" included all individuals with the ICD10 D51 diagnosis as cases. FinnGen summary statistics included prefiltered variants (minimum allele count >5 and imputation INFO score >0.6). For more information on genotype data, disease endpoints and GWAS analyses, please see https://finngen.gitbook.io/documentation/.

*GWAS meta-analysis.* We extracted all genetic variants with a rs-number from the summary statistics of the three participating cohorts and conducted an inverse variance weighted fixed-effects meta-analysis without genomic control using GWAMA (v2.2.2)[13]. The genomic inflation factors (lambda) of the individual study summary statistics were 0.69 (UKBB), 0.92 (EstBB), and 1.05 (FinnGen). The low lambda value in the UKBB dataset can be attributed to low allele counts for relatively rare variants in this sample with an unbalanced case:control ratio, which leads to deflation in the bottom left corner in the QQ plot[12]. A total of 30,907,385 variants were included in the meta-analysis (meta-analysis lambda calculated for variants present in all three cohorts—1.02). Genome-wide significance was set to *p* < 5 × 10⁻⁸. All the reported lead signals had imputation INFO scores between 0.96–1 in EstBB and UKBB and >0.6 in FinnGen (Supplementary Data 1).

We also performed sex-stratified meta-analysis for the five lead signals, using data from EstBB and UKBB. Effect heterogeneity between sexes was evaluated using GWAMA gender heterogeneity *p* value[13].

*Finemapping.* For finemapping, the 1MB locus around the GWAS lead SNP was analysed by standard approximate Bayesian finemapping approach as implemented into CRAN R package, *corrcoverage* v1.2.1 (https://annahutch.github.io/corrcoverage/index.html)[14,15]. The ppfunc function was used to convert the marginal *Z*-scores to posterior probabilities of causality, using default prior for the standard deviation of the effect size parameter (W = 0.2). The *Z*-scores were previously calculated by dividing association ln(OR) with standard error of ln(OR) and SNPs with MAF >0.001 were included to the analysis. Received 95% credible set SNPs were plotted against 15-state chromatin segmentations for 127 samples from the Roadmap Epigenomics project (http://egg2.wustl.edu/roadmap/data/byFileType/chromhmmSegmentations/ChmmModels/coreMarks/jointModel/final/) and figure fine-tuning was done using Inkscape 1.1.0-dev (0486c1a, 2020-10-10) (https://inkscape.org/).

*HLA allele imputation in the EstBB.* Imputation of HLA alleles from SNP data was carried out using the SNP2HLA tool[41]. As an imputation reference we used a merged reference of EstBB WGS samples[40] and Type 1 Diabetes Genetics Consortium reference[41]. The reported associated alleles (HLA-DQB1*06:02, HLA-DQA1*01:02, and HLA-DRB1*15:01) all had imputation INFO scores >0.98. Conditional analysis for the reported HLA alleles and the HLA region lead variant (rs4148874) in the EstBB data was done using SAIGE[12].

*Colocalization.* We conducted colocalization analyses to detect shared causal variants between pernicious anemia and gene expression using COLOC (v.3.2.1) R package[16] and GWAS meta-analysis summary statistics. We set the prior probabilities to $p_1 = 1 \times 10^{-4}$, $p_2 = 1 \times 10^{-4}$, $p_{12} = 5 \times 10^{-6}$ as suggested by Wallace (2020)[42] and used the COLOC version which takes regression coefficients and their variance into account.

In the analysis we compared our significant GWAS loci to all eQTL Catalog[43] (https://www.ebi.ac.uk/eqtl/) RNA-seq datasets (excluding Lepik et al. 2017[44] due to

sample overlap) containing QTLs for gene expression, exon expression, transcript usage, and txrevise event usage; eQTL Catalog microarray datasets containing QTLs for gene expression (excluding Kasela et al. 2017[45]); and GTEx v8 datasets containing QTLs for gene expression (see Methods: https://www.ebi.ac.uk/eqtl/Methods/).

We lifted the GWAS summary statistics over to hg38 build to match the eQTL Catalog. For each genome-wide significant ($p < 5 \times 10^{-8}$) GWAS locus we extracted the 1 Mbp radius of its top hit from QTL datasets and ran the colocalization analysis for those eQTL Catalog traits that had at least one cis-QTL within this region with $p < 1 \times 10^{-6}$. We considered two signals to colocalize if the posterior probability for a shared causal variant was 0.8 or higher. All results with a PP4 > 0.8 can be found in Supplementary Data 2.

Results were visualized with the *LocusCompareR* library (v1.0.0)[46].

*Look-up of phenome-wide associations in GWAS catalog and with PhenoScanner v2*. FUMA v1.3.6a[47] was used to compare the genome-wide significant lead signals and markers in high LD with these markers against the results in the GWAS catalog. The results of this look-up are presented in Supplementary Data 2.

PhenoScanner v2[48,49] was used for look-up of phenotype associations for the GWAS lead variants in previous GWAS studies. PhenoScanner query was done using the rsid-s of GWAS lead variants and the *phenoscanner* R package (https://github.com/phenoscanner/phenoscanner). Query results were filtered to keep one association per variant per trait, keeping studies from newer or larger studies. Descriptions of experimental factor ontology (EFO) terms and classification of EFO broad categories were obtained from the GWAS Catalog. Missing categories were added by manually searching the EMBL-EBI EFO webpage (www.ebi.ac.uk/efo/). For visualization of PhenoScanner results, parent categories with fewer results were grouped into larger categories and a heatmap was created using the *pheatmap* library in R 3.6.1. and a modified script from (https://github.com/LappalainenLab/spiromics-covid19-eqtl/blob/master/eqtl/summary_phenoscanner_lookup.Rmd). The results of this look-up are presented in Supplementary Data 3.

*Mouse phenotypes*. We used the Mouse Genome Database[17] (http://www.informatics.jax.org) to evaluate the *PNPT1* effect on phenotype in mouse models. We downloaded the data on MCV in *Pnpt1tm1a(KOMP)Wtsi* mutant mice (Supplementary Data 4) from (https://www.mousephenotype.org/data/charts?accession=MGI:1918951&allele_accession_id=MGI:4364657&pipeline_stable_id=M-G-P_001&procedure_stable_id=M-G-P_016_001¶meter_stable_id=M-G-P_016_001_005&zygosity=heterozygote&phenotyping_center=WTSI).

*Look-up of variants associated with relevant phenotypes*. We conducted a look-up of variants associated with relevant phenotypes (vitamin B12 levels and gastric parietal cell autoantibody positivity) in our GWAS meta-analysis summary statistics. We used a list of variants reported in association with vitamin B12 levels in the general population (Icelandic and Danish data) by Grarup et al.[8] and variants associated with parietal cell autoantibody positivity in type 1 diabetes patients[9,24] (Supplementary Data 5).

We further conducted a look-up for variants associated with vitiligo and (autoimmune) thyroid issues, conditions commonly co-occurring with pernicious anemia, to rule out the confounding effect of these concomitant diagnoses. For vitiligo, we chose variants associated with melanocyte biology[50] and for thyroid issues (Graves' disease and Hashimoto thyreoiditis) we selected variants that do not have an obvious role in the immune system regulation from the GWAS catalog and from a study by Cooper et al.[30] (Supplementary Data 5).

*Analysis of associated phenotypes in EstBB*. Using the individual level data in the EstBB, we conducted an analysis to find ICD10 diagnosis codes associated with the D51.0 diagnosis. We tested the association between pernicious anemia status (defined as ICD10 D51.0) and other ICD10 codes using logistic regression and adjusting for sex, age, and ten PCs. Bonferroni correction was applied to select statistically significant associations (Number of tested ICD main codes—1944, corrected $p$ value threshold—$2.5 \times 10^{-5}$). Results were visualized using the *PheWas* library (https://github.com/PheWAS/PheWAS). All analyses were carried out in R 3.6.1. The results of this analysis are presented in Supplementary Data 6.

*Analysis of autoimmune disease prevalence*. To test the prevalence of other autoimmune diseases in pernicious anemia cases, we made a list of 40 autoimmune diagnoses[51] (Supplementary Data 7) and checked their cumulative prevalence (% of individuals having at least one of these diagnoses) among the EstBB and UKBB cohorts, for which we had access to individual level data.

**Reporting Summary**. Further information on research design is available in the Nature Research Reporting Summary linked to this article.

## Data availability

Used UKBB and FinnGen summary statistics can be browsed and downloaded from UKBB PheWeb (http://pheweb.sph.umich.edu/SAIGE-UKB/) and FinnGen PheWeb (http://r3.finngen.fi), respectively. Full meta-analysis summary statistics can be downloaded from http://www.geenivaramu.ee/tools/pernicious_anemia_Laisketal2021_sumstats.gz. All GWAS analyses and meta-analysis were carried out with standard tools and pipelines. The analyses in

this paper also use data from the Mouse Genome Database: http://www.informatics.jax.org; International Mouse Phenotyping Consortium: https://www.mousephenotype.org; GTEx Portal: https://gtexportal.org/home/; eQTL Catalog: https://www.ebi.ac.uk/eqtl/; GWAS Catalog: https://www.ebi.ac.uk/gwas/; Roadmap Epigenomics project (http://egg2.wustl.edu/roadmap/data/byFileType/chromhmmSegmentations/ChmmModels/coreMarks/jointModel/final/).

## Code availability

In EstBB, GenomeStudio (v2.0.4), Eagle (v2.3), and Beagle (v28Sep18.793) were used as part of the standard genotyping and imputation pipeline. Cohort-level analyses were carried out with SAIGE (v0.38). HLA allele imputation was carried out with SNP2HLA 1.0.3. Central meta-analysis was conducted using the GWAMA software (v2.2.2). Finemapping was carried out with R package *corrcoverage* v1.2.1 (https://annahutch.github.io/corrcoverage/index.html). Inkscape 1.1.0-dev (0486c1a, 2020-10-10) was used for finetuning the figures. For colocalization, COLOC (v3.2.1) was used and results were visualized with *LocusCompareR* (v1.0.0) library. FUMA v1.3.6a was used for GWAS catalog (e91_r2018-02-06) look-up. PhenoScanner v2 was used for look-up of phenotype associations for the GWAS lead variants in previous GWAS studies, using the *phenoscanner* (v1.0) R package, and the results were visualized using *pheatmap* library in R 3.6.1. and a modified script from (https://github.com/LappalainenLab/spiromics-covid19-eqtl/blob/master/eqtl/summary_phenoscanner_lookup.Rmd). Associated phenotypes analysis was visualized with the *PheWas* library (0.99.5–4) (https://github.com/PheWAS/PheWAS). All other analyses were conducted in R 3.6.1.

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

# ARTICLE

21. Ungar, B., Mathews, J. D., Tait, B. D. & Cowling, D. C. HLA-DR patterns in pernicious anaemia. *Br. Med. J.* **282**, 768–770 (1981).
22. Bruserud, Ø., Oftedal, B. E., Wolff, A. B. & Husebye, E. S. AIRE-mutations and autoimmune disease. *Curr. Opin. Immunol.* **43**, 8–15 (2016).
23. Yang, S., Bansal, K., Lopes, J., Benoist, C. & Mathis, D. Aire's plant homeodomain(PHD)-2 is critical for induction of immunological tolerance. *Proc. Natl Acad. Sci. USA* **110**, 1833–1838 (2013).
24. Plagnol, V. et al. Genome-wide association analysis of autoantibody positivity in type 1 diabetes cases. *PLoS Genet.* **7**, e1002216 (2011).
25. Goerss, J. B. et al. Risk of fractures in patients with pernicious anemia. *J. Bone Miner. Res.* **7**, 573–579 (1992).
26. Ye, W. & Nyrén, O. Risk of cancers of the oesophagus and stomach by histology or subsite patients hospitalised for pernicious anaemia. *Gut* **52**, 938–941 (2003).
27. Bennett, M. Vitamin B12 deficiency, infertility and recurrent fetal loss. *J. Reprod. Med. Obstet. Gynecol.* **46**, 209–212 (2001).
28. Reznikoff-Etiévant, M. F., Zittoun, J., Vaylet, C., Pernet, P. & Milliez, J. Low vitamin B12 level as a risk factor for very early recurrent abortion. *Eur. J. Obstet. Gynecol. Reprod. Biol.* **104**, 156–159 (2002).
29. Hübner, U. et al. Low serum vitamin B12 is associated with recurrent pregnancy loss in Syrian women. *Clin. Chem. Lab. Med.* **46**, 1265–1269 (2008).
30. Cooper, J. D. et al. Seven newly identified loci for autoimmune thyroid disease. *Hum. Mol. Genet.* **21**, 5202–5208 (2012).
31. Dhir, A. et al. Mitochondrial double-stranded RNA triggers antiviral signalling in humans. *Nature* **560**, 238–242 (2018).
32. Lee-Kirsch, M. A. The type I interferonopathies. *Annu. Rev. Med.* **68**, 297–315 (2017).
33. Stefan-Lifshitz, M. et al. Epigenetic modulation of β cells by interferon-α via PNPT1/mir-26a/TET2 triggers autoimmune diabetes. *JCI Insight* **4**, e126663 (2019).
34. Cong, B., Zhang, Q. & Cao, X. The function and regulation of TET2 in innate immunity and inflammation. *Protein Cell* **12**, 165–173 (2021).
35. Chen, M. H. et al. Trans-ethnic and ancestry-specific blood-cell genetics in 746,667 individuals from 5 global populations. *Cell* **182**, 1198–1213.e14 (2020).
36. Eriksson, D. et al. GWAS for autoimmune Addison's disease identifies multiple risk loci and highlights AIRE in disease susceptibility. *Nat. Commun.* **12**, 959 (2021).
37. Cotsapas, C. & Hafler, D. A. Immune-mediated disease genetics: the shared basis of pathogenesis. *Trends Immunol.* **34**, 22–26 (2013).
38. Loh, P. R. et al. Reference-based phasing using the haplotype reference consortium panel. *Nat. Genet.* **48**, 1443–1448 (2016).
39. Browning, S. R. & Browning, B. L. Rapid and accurate haplotype phasing and missing-data inference for whole-genome association studies by use of localized haplotype clustering. *Am. J. Hum. Genet.* **81**, 1084–1097 (2007).
40. Mitt, M. et al. Improved imputation accuracy of rare and low-frequency variants using population-specific high-coverage WGS-based imputation reference panel. *Eur. J. Hum. Genet.* **25**, 869–876 (2017).
41. Jia, X. et al. Imputing amino acid polymorphisms in human leukocyte antigens. *PLoS ONE* **8**, e64683 (2013).
42. Wallace, C. Eliciting priors and relaxing the single causal variant assumption in colocalisation analyses. *PLoS Genet.* **16**, e1008720 (2020).
43. Kerimov, N. et al. eQTL catalogue: a compendium of uniformly processed human gene expression and splicing QTLs. Preprint at *bioRxiv* https://doi.org/10.1101/2020.01.29.924266 (2020).
44. Lepik, K. et al. C-reactive protein upregulates the whole blood expression of CD59 - an integrative analysis. *PLoS Comput. Biol.* **13**, e1005766 (2017).
45. Kasela, S. et al. Pathogenic implications for autoimmune mechanisms derived by comparative eQTL analysis of CD4 + versus CD8 + T cells. *PLoS Genet.* **13**, e1006643 (2017).
46. Liu, B., Gloudemans, M. J., Rao, A. S., Ingelsson, E. & Montgomery, S. B. Abundant associations with gene expression complicate GWAS follow-up. *Nat. Genet.* **51**, 768–769 (2019).
47. Watanabe, K., Taskesen, E., van Bochoven, A. & Posthuma, D. Functional mapping and annotation of genetic associations with FUMA. *Nat. Commun.* **8**, 1826 (2017).
48. Staley, J. R. et al. PhenoScanner: a database of human genotype-phenotype associations. *Bioinformatics* **32**, 3207–3209 (2016).
49. Kamat, M. A. et al. PhenoScanner V2: an expanded tool for searching human genotype-phenotype associations. *Bioinformatics* **35**, 4851–4853 (2019).
50. Jin, Y. et al. Genome-wide association studies of autoimmune vitiligo identify 23 new risk loci and highlight key pathways and regulatory variants. *Nat. Genet.* **48**, 1418–1424 (2016).
51. Harpsøe, M. C. et al. Body mass index and risk of autoimmune diseases: a study within the Danish National Birth Cohort. *Int. J. Epidemiol.* **43**, 843–855 (2014).

## Acknowledgements
This study is supported by the Estonian Research Council grants MOBTP155 and PRG687. Computations were performed in the High Performance Computing Center, University of Tartu. This study has been conducted using the UK Biobank Resource under Application Number 17085. We want to acknowledge the participants and investigators of the FinnGen and UKBB studies. The Genotype-Tissue Expression (GTEx) Project was supported by the Common Fund of the Office of the Director of the National Institutes of Health, and by NCI, NHGRI, NHLBI, NIDA, NIMH, and NINDS. The data used for the analyses described in this manuscript were obtained from the GTEx Portal on 10/01/20.

## Author contributions
T.L., M.L., M.K., and R.M. carried out data analysis and interpreted the results. E.A. interpreted the results. E.B.R.T. prepared and provided the data for EstBB. All authors contributed to writing the manuscript.

## Competing interests
The authors declare no competing interests.

## Additional information

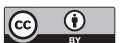

## Estonian Biobank Research Team
Andres Metspalu[1], Mari Nelis[1], Lili Milani[1] & Tõnu Esko[1]

