## [Peer Review File · Nature Communications]

REVIEWER COMMENTS

Reviewer #1 (Remarks to the Author):

Comments to the authors:

Laisk, Lepamets, and Mägi carried out a large GWAS study designed to identify genetic variants associated with pernicious anemia. The authors took advantage of a number of large and publicly available datasets and also used new dataset derived from Estonian Biobank. The data are presented as results from each of the biobanks, Estonian Biobank, UK Biobank, and the FinnGen study. A “meta analysis” employing all studies was also carried out.

The authors report the identification of several variants that met genome wide association level of statistical significance. The majority of these are biologically plausible as they are involved in the immune response and autoimmunity to gastric cells producing intrinsic factor is a known cause of pernicious anemia. That said, the authors do little explore alternatives to the most obvious candidate in any given region. This is an error that others have made in the past. There have been other instances when genes other than the obvious candidate was later found to be the source of the real signal.

The authors make excellent use of existing data. For 4 of the 5 lead SNPs the effect size was very similar between the three samples. This may be because all of the samples were derived from population of European ancestry and from countries with roughly comparable medical systems and environmental conditions.

Vitamin B12 deficiency (usually from malabsorption) is the most common underlying cause of pernicious anemia. It is difficult to diagnose without multiple biochemical tests. The authors have bypassed this complexity by relying on ICD10 codes. This is a strength in that it simplifies the data analysis and a weakness because it is likely to include other cause of pernicious anemia.. It is also a noisy variable as the use of the code may vary between and within studies. As such, one would expect there are many additional genetic associations that are “real” but did not reach genome wide significance.

The strict reliance on the codes may also explain there were so few cases in these large cohorts where one would expect a higher frequency of vitamin B12 deficiency. There may be many more cases that were not coded with frank anemia because the patients received treatment with oral or injectable vitamin B12. The authors should include some discussion about how undercounting would impact their results.

Reviewer #2 (Remarks to the Author):

In this paper the authors use data from three national biobanks to study the genetics of pernicious anaemia (PA). They claim five loci, all of which have been impacted in other autoimmune diseases. PA has not been subjected to many well powered large scale genetic studies in the past and thus this paper does represent a step forward. The results appear a bit limited as it was unclear whether the affected individuals in all the studies suffered from other autoimmune diseases which might confuse

the results. In addition, I did not think that the results were considered in the full setting of known data. This is not my exact field, but I managed to identify a few uncited relevant papers. Some specific points are listed below.

Figure 1. Cannot link the data in figure 1C to the Manhattan plot, this should be corrected.

As far as I can tell the genotyping chips and the imputation strategies differed in all three cohorts studied. Why did the authors not start with genotyped dataset and then impute with common sets of reference data?

What is the relationship between the PA risk SNP rs6679677 and the risk alleles of other autoimmune diseases associated with PTPN22? Similarly, the IL2RA locus has been associated with several autoimmune disease including T1D, is this signal shared with PA?

Establishing whether the cases with PA have other autoimmune diseases is important given the shared associations described. Why did the authors only seek this information in the Estonian biobank? Did they formerly check for associations with, for example thyroid disease, to ensure that the PA signal was not driven by thyroid disease?

The association AIRE prompts the question: is this signal is present across the PA cohorts or enriched in those with multiple autoimmune diseases?

I found this paper looking at PA GWAS in Chinese cohort which was not cited: Human Molecular Genetics, Volume 21, Issue 11, 1 June 2012, Pages 2610–2617

There is quite an extensive literature on genetics and serum vitamin B12 levels, this was not mentioned or compared.

I also found a paper looking at the genetics of anti-GPC production, which is highly relevant: <https://doi.org/10.1371/journal.pgen.1002216>

Reviewer #3 (Remarks to the Author):

Laisk T et al did a meta-GWAS analysis of pernicious anemia using the data from three independent cohorts of European origin. The methods are robust, and novel loci have been discovered for pernicious anemia with solid evidence.

While I found the study to be interesting, I believe that the study and paper can benefit from addressing the following issues/concerns:

1. 1) Proportion of cases seems to be 2-3 fold higher in the FinnGen study. Is this because of a different inclusion criteria or is prevalence known to be higher among Finns?
2. A different number of PCs were included for adjustment in the regression analysis of each cohort. How was the number of PCs determined for each cohort? Good to report the lambda gc values for the GWAS results of individual cohorts and the meta-GWAS analysis, and discuss potential impact of highly imbalanced case-control proportions on measuring the impact of population stratification.

3. How were imputed genotypes QCed? How was the association test performed for imputed genotypes (using dosage)? As for the leading SNPs, are they imputed SNPs? If yes, what are the quality scores of these imputed SNPs? And, were the associations within these loci supported by any genotyped SNPs?
4. As for the leading variant on Chr 2 locus, why was rs12616502 absent in the Finnish data? are there any supporting evidences from other variants within the locus in the Finnish Data?
5. Would be good to present the individual association data for these SNPs in the 3 datasets together with the meta-analysis data. The results of genetic heterogeneity test should be provided for all the loci (either in the table 1 or supplementary table). Some of the reported signals are marginally beyond genome-wide thresholds (at chr 2, 10, 21) and it would be important to understand if any these associations were driven by a particular study.
6. For the colocalization analysis of GWAS and eQTL association effects, it stated that “pernicious anemia has a common causal variant with the expression of PTPN1 in thyroid tissue in GTEx v8 dataset.”. how was “the causal variant” determined?
7. What would be the relevance for coloc analysis with thyroid tissue for the chr2 variant associated with B12 deficiency anemia? Would evaluations using blood cell data or gut tissue data be more relevant? Although PNPT1 shows strong coloc posterior probability with expression in the thyroid, equally strong or stronger signals are also seen for CCDC104 in blood cell lines and fat.
8. It will be also helpful to show and comment on the results of colocalization analysis for other loci.
9. Are the results at the three HLA alleles fully correlated? It will be helpful to provide the result for the HLA-DR15 haplotype. Will conditioning on the effects of the HLA alleles or HLA-DR15 abolish the association signals within the MHC region?
10. The look-up of pernicious anemia with other traits/diseases in the Estonian biobank data is interesting. Were any specific SNPs identified from the study associated with these traits/diseases? Could these associations be evaluated in a Mendelian randomization or a Mediation framework to evaluate potential causal path of pernicious anemia with other diseases?
11. As for the “Time At First Diagnosis” analysis, all the results are not statistically significant, and the results could be just “chance events”.
12. Several GWAS has been performed for B12 deficiency (such as Lin et al. Hum Mol Genet 2012) and B12 serum level (such as Grarup et al PLoS Genetics 2013). Should perform a systematic comparison of the results from the current studies with the ones of these previous studies.
13. It will be good to include the rsid of lead SNPs in the Manhattan plot on fig1 and the description of ICD codes in Fig3b for easy reference.

Reviewer #1 (Remarks to the Author):

Comments to the authors:

Laisk, Lepamets, and Mägi carried out a large GWAS study designed to identify genetic variants associated with pernicious anemia. The authors took advantage of a number of large and publicly available datasets and also used new dataset derived from Estonian Biobank. The data are presented as results from each of the biobanks, Estonian Biobank, UK Biobank, and the FinnGen study. A “meta analysis” employing all studies was also carried out.

The authors report the identification of several variants that met genome wide association level of statistical significance. The majority of these are biologically plausible as they are involved in the immune response and autoimmunity to gastric cells producing intrinsic factor is a known cause of pernicious anemia. That said, the authors do little explore alternatives to the most obvious candidate in any given region. This is an error that others have made in the past. There have been other instances when genes other than the obvious candidate was later found to be the source of the real signal.

The authors make excellent use of existing data. For 4 of the 5 lead SNPs the effect size was very similar between the three samples. This may be because all of the samples were derived from population of European ancestry and from countries with roughly comparable medical systems and environmental conditions.

Vitamin B12 deficiency (usually from malabsorption) is the most common underlying cause of pernicious anemia. It is difficult to diagnose without multiple biochemical tests. The authors have bypassed this complexity by relying on ICD10 codes. This is a strength in that it simplifies the data analysis and a weakness because it is likely to include other cause of pernicious anemia.. It is also a noisy variable as the use of the code may vary between and within studies. As such, one would expect there are many additional genetic associations that are “real” but did not reach genome wide significance.

The strict reliance on the codes may also explain there were so few cases in these large cohorts where one would expect a higher frequency of vitamin B12 deficiency. There may be many more cases that were not coded with frank anemia because the patients received treatment with oral or injectable vitamin B12. The authors should include some discussion about how undercounting would impact their results.

Thank you for this positive feedback! We agree that mapping potential causal genes in GWAS studies is a challenging task and relying solely on the most “obvious” candidate genes in the region can be a slippery slope. Therefore we adopted a more data-driven approach and use finemapping and colocalisation analyses together with data from mouse knockouts to select the credible set of causal SNPs in each locus and consecutively the most likely causal gene. The credible sets of most likely causal SNPs at each associated locus were determined using the standard approximate Bayesian finemapping approach. When selecting the most likely candidate gene in each associated region, we considered the following criteria – a) whether the credible set includes a coding variant in any of the nearby genes, b) whether the signal colocalises with a variant that affects gene

expression in COLOC analysis, and c) relevant biological functions of the neighboring genes and the mouse phenotypes of corresponding gene knock-outs. We have now added this information to the manuscript as well, both to the results and methodology section, and we hope this provides sufficient evidence for each proposed candidate gene.

As to the other point raised in the review (mainly the use of ICD codes to identify cases), we agree the use of the D51.0 ICD-10 code for identifying pernicious anemia cases may differ in UK, Estonia and Finland, and furthermore, the subset of cases (and the severity of their disease) we include in our analysis also depends on where the biobanks get their information (mainly hospital data in UKBB and Finland, and primary care + hospital data in Estonia). That been said, the prevalence of pernicious anemia in our studied datasets ranged from 0.2-0.8%, which is roughly in line with the expected prevalence of pernicious anemia (0.1% in the general population and >2% in over 60-year-olds), given that these analysed datasets include individuals from a very broad age range. Here it is important to distinguish between vitamin B12 deficiency (often caused by diet) and pernicious anemia, a subtype of B12 deficiency most often caused by autoimmune destruction of gastric parietal cells and concomitant intrinsic factor deficiency, leading to impaired B12 absorption and vitamin deficiency. Since we see no significant differences in effect estimates for the reported lead variants, and the identified loci and candidate genes have a clear role in autoimmune regulation, we believe the vast majority of cases in this study indeed have autoimmune B12 deficiency (pernicious anemia), and potential differences between disease classification has not caused major heterogeneity in the analysed data. In worst case scenario, the potential misclassification of our control subjects as not having pernicious anemia can increase the heterogeneity in the analysed data and attenuate the results towards the null, meaning that either larger datasets with the current phenotype definition or further refinement of the phenotype definition is needed to increase the number of identified loci. We have now added these discussion points in the manuscript Discussion (page 8 of the track changes version).

Reviewer #2 (Remarks to the Author):

In this paper the authors use data from three national biobanks to study the genetics of pernicious anaemia (PA). They claim five loci, all of which have been impacted in other autoimmune diseases. PA has not been subjected to many well powered large scale genetic studies in the past and thus this paper does represent a step forward. The results appear a bit limited as it was unclear whether the affected individuals in all the studies suffered from other autoimmune diseases which might confuse the results. In addition, I did not think that the results were considered in the full setting of known data. This is not my exact field, but I managed to identify a few uncited relevant papers. Some specific points are listed below.

Figure 1. Cannot link the data in figure 1C to the Manhattan plot, this should be corrected.

We have now annotated the Manhattan plot with rs-numbers at association peaks to link with Figure 1C.

As far as I can tell the genotyping chips and the imputation strategies differed in all three cohorts studied. Why did the authors not start with genotyped dataset and then impute with common sets of reference data?

Thank you for this question. Indeed, imputing included cohort data with common reference data has been a commonly used approach in genetic association studies. However, when it comes to large-scale population-based biobank data (here over 650,000 participants) this approach is somewhat complicated due to restrictions of accessing individual-level genotype data, the pure volume of such computational task and legal restrictions which do not allow to upload individual level data to the commonly used Haplotype Reference Consortium imputation server, for example. Moreover, two of the included datasets have been imputed with population-specific references, which improves imputation quality. Therefore we decided to use the already existing summary statistics level data for UKBB (imputed to the Haplotype Reference Consortium panel) and FinnGen (imputed to SISu v3 reference panel) and in the EstBB, conducted the association analysis using our existing imputed data (imputed to the Estonian specific reference; PMID: 28401899).

What is the relationship between the PA risk SNP rs6679677 and the risk alleles of other autoimmune diseases associated with PTPN22? Similarly, the IL2RA locus has been associated with several autoimmune disease including T1D, is this signal shared with PA?

rs6679677 PA risk increasing allele is A. In Phenoscanner and GWAS catalog look-up, the A allele in this locus was also associated with an increased risk of systemic lupus erythematosus, T1D, rheumatoid arthritis and other autoimmune diseases, so we see consistent effects across a wide range of autoimmune conditions. rs2476491 near IL2RA (PA risk increasing allele A) was associated with an increased risk of hayfever/allergic rhinitis/eczema and self-reported hypothyroidism (and SNPs in LD are associated additionally with vitiligo, autoimmune thyroid disease/hypothyroidism and multiple sclerosis), so this locus is distinct from the known T1D associations.

Establishing whether the cases with PA have other autoimmune diseases is important given the shared associations described. Why did the authors only seek this information in the Estonian biobank? Did they formerly check for associations with, for example thyroid disease, to ensure that the PA signal was not driven by thyroid disease?

We would like to thank the reviewer for pointing out this interesting question. We have access to individual level data of the EstBB and UKBB, therefore we conducted the lookup in this dataset. To establish whether pernicious anemia cases have other autoimmune disease, we compiled a list of 40 most common autoimmunity-related diagnoses and tested the prevalence of these autoimmune diseases among the pernicious anemia cases compared to controls. In EstBB, 55.8% of all pernicious anemia cases have at least one other autoimmune diagnosis (23.9% in controls). For comparison, we did a similar look-up for other common autoimmune diseases as well – type 1 diabetes (35.9% in cases vs 22.8% in controls), vitiligo (37.9% vs 23.5%), rheumatoid arthritis (39.3% vs 19.2%) and

Hashimoto's thyroiditis (33.7% vs 18.6%). This confirms the higher incidence of autoimmune diseases in pernicious anemia as well as other autoimmune diseases. To check whether the pernicious anemia genetic associations were driven by concomitant autoimmune diseases (mostly vitiligo and thyroid problems, which were also highlighted as significant in the associated diagnoses analysis), we did a look-up for vitiligo, hypothyroidism and autoimmune thyroid disease associations in our meta-analysis summary statistics (Supplementary Data 5). Since autoimmune diseases can share pathogenic mechanisms, we focused on loci that according to current knowledge do not regulate autoimmune response. For vitiligo, we chose variants annotated to genes associated with melanocyte biology and for thyroid issues we chose the TSHR (thyroid stimulating hormone receptor) and TPO (thyroid peroxidase) loci and others (Supplementary Data 5). None of these variants reached a nominal significance in our pernicious anemia GWAS meta-analysis, confirming that the observed associations are not mainly driven by concomitant autoimmune disease.

It is known that autoimmune diseases share causal genetic mechanisms and often cluster together, and loci associated with one autoimmune disease are often pleiotropic and also affect the risk of other autoimmune diseases.

The association AIRE prompts the question: is this signal is present across the PA cohorts or enriched in those with multiple autoimmune diseases?

The association with AIRE missense variant rs74203920 was observed in all three analysed cohorts, with relatively similar effect estimates (Figure 1C and Table 1; heterogeneity p-value 0.05). In order to evaluate the phenotypic effect of the AIRE missense variant rs74203920, we conducted a pheWAS analysis for the alternative allele carrier status. In the EstBB dataset, 4,882 individuals were either heterozygous or homozygous for the alternative allele. The only diagnosis group showing significantly increased prevalence in the alternative allele carriers was D51 for vitamin B12 deficiency anemias (including D51.0 for pernicious anemia). The phenotypic effect of rs74203920 has not been described before, but on a molecular level, the amino acid change could lead to a change in binding partners or affects binding of the stabilizing Zn²⁺ molecule [PMID: 23319629]. We have now added this information to the manuscript as well.

I found this paper looking at PA GWAS in Chinese cohort which was not cited: Human Molecular Genetics, Volume 21, Issue 11, 1 June 2012, Pages 2610–2617

There is quite an extensive literature on genetics and serum vitamin B12 levels, this was not mentioned or compared.

I also found a paper looking at the genetics of anti-GPC production, which is highly relevant: <https://doi.org/10.1371/journal.pgen.1002216>

We thank the reviewer for pointing out these papers. We have now conducted a systematic comparison between our results, a large study evaluating genetic influencers of B12 levels in European ancestry subjects, and a study evaluating genetic determinants of parietal cell antibody (PCA) positivity in T1D patients. In short, we find none of the variants influencing B12 levels in the general population were genome-wide significant in our data, but 8/11 showed nominal association p-values, with the allele decreasing serum B12 levels consistently also increasing

the risk of pernicious anemia. Similarly, six of the nine T1D risk loci showing significant PCA associations were nominally significant in our meta-analysis (Supplementary Data 5), including rs2476601 (PTPN22), which is one of our top associated variants. The associated loci have a central role in immune regulation (incl PTPN22, CTLA4, IFIH1, HLA region and SH2B3). Notably, we did not see an association with the parietal cell autoantibody-associated INS locus, which again suggests that the genome-wide significant loci we report are not confounded by accompanying autoimmune disease (such as T1D where INS plays a central role in the etiopathogenesis).

Reviewer #3 (Remarks to the Author):

Laisk T et al did a meta-GWAS analysis of pernicious anemia using the data from three independent cohorts of European origin. The methods are robust, and novel loci have been discovered for pernicious anemia with solid evidence.

While I found the study to be interesting, I believe that the study and paper can benefit from addressing the following issues/concerns:

1. Proportion of cases seems to be 2-3 fold higher in the FinnGen study. Is this because of a different inclusion criteria or is prevalence known to be higher among Finns?

As far as we know, there is no comprehensive overview of the exact population prevalence in different countries, but according to this study (<https://care.diabetesjournals.org/content/43/5/1041>), the prevalence of atrophic gastritis is 0.2% in a group of non-diabetic individuals in Finland, which overlaps with the estimated population prevalence of pernicious anemia (0.1%-2%). Therefore it is likely that since the FinnGen study data comes from a more broader definition (D51, all vitamin B12 deficiency anemias), this may play a role. Additionally, it is possible the Finnish cases are older (the prevalence of pernicious anemia increases with age), but since we do not have access to individual level data, we cannot test this.

2. A different number of PCs were included for adjustment in the regression analysis of each cohort. How was the number of PCs determined for each cohort? Good to report the lambda gc values for the GWAS results of individual cohorts and the meta-GWAS analysis, and discuss potential impact of highly imbalanced case-control proportions on measuring the impact of population stratification.

The lambda values for the individual cohorts were as follows: UKBB - 0.69, Estbb - 0.92, and FinnGen - 1.05. Population stratification in each cohort was accounted for during the SAIGE analysis, which uses a mixed model for association testing and is especially well-suited for phenotypes with a pronounced case-control imbalance (such as pernicious anemia). SAIGE developers have suggested (<https://github.com/weizhouUMICH/SAIGE/issues/196>) to also add PCs as covariates to potentially capture any residual stratification (and it has been shown that inclusion of PCs may also speed up the calculations). The number of PCs is determined by the default association testing pipeline in each cohort.

3. How were imputed genotypes QCed? How was the association test performed for imputed genotypes (using dosage)? As for the leading SNPs, are they imputed SNPs? If yes, what are the quality scores of these imputed SNPs? And, were the associations within these loci supported by any genotyped SNPs?

In the FinnGen pipelines, the following filters were applied: minimum allele count 5 and imputation INFO >0.6. For the UKBB, the summary statistics files have been pre-filtered to include only variants with an INFO score ≥ 0.3 . The imputation quality scores for each variant have now been shown in Supplementary Table 1, and all the top variants have INFO scores (0.96-1). For the UKBB and FinnGen we used publicly available GWAS summary statistics (we did not carry out the analysis testing ourselves and do not have access to individual level genotype data), and in the EstBB we analysed the data ourselves, using dosages for association testing. In the EstBB, all variants are imputed as the imputation pipeline does not distinguish between genotyped and imputed variants in the output.

4. As for the leading variant on Chr 2 locus, why was rs12616502 absent in the Finnish data? are there any supporting evidences from other variants within the locus in the Finnish Data?

Unfortunately we do not know why this variant is not present in the FinnGen dataset, but since this variant does exist in the Finnish population, this is probably due to post-genotyping or post-imputation filtering. In this region, the credible set included 17 variants, including two that most likely have functional consequences via affecting gene expression (rs7586115 and rs13420929, $r=0.7$ with lead signal rs12616502). Unlike the lead signal, these two variants are also present in the FinnGen dataset, although statistically not significant (in FinnGen data, rs7586115 $p=0.51$, OR 1.07 (0.87-1.33); meta-analysis heterogeneity p -value 0.01). It is difficult to hypothesise with current data why exactly this variant does not have a similar effect on the phenotype in the Finnish data - it could for example be due to potential differences in the phenotype, or attributable to the specific genetic structure of the Finnish population. The effect estimate in the current analysis also does not rule out an effect on the phenotype, as 95% confidence interval indicates an OR between 0.87-1.33, therefore it can also be a power issue.

5. Would be good to present the individual association data for these SNPs in the 3 datasets together with the meta-analysis data. The results of genetic heterogeneity test should be provided for all the loci (either in the table 1 or supplementary table). Some of the reported signals are marginally beyond genome-wide thresholds (at chr 2, 10, 21) and it would be important to understand if any these associations were driven by a particular study.

We have now added the heterogeneity p -values to Table 1 and the statistics from individual studies to Supplementary Data 1. As can be seen from Table 1 and Figure 1C, the effect estimates are comparable in different cohorts and results are not driven by a single study (except for the locus on chromosome 2, where the lead signal is missing in FinnGen data and likely causal variants have a heterogeneity p -value of 0.01; please see the answer above).

6. For the colocalization analysis of GWAS and eQTL association effects, it stated that “pernicious anemia has a common causal variant with the expression of PTPN1 in thyroid tissue in GTEx v8 dataset.”. how was “the causal variant” determined?

We have now rephrased the sentence to: “Colocalization analysis showed pernicious anemia GWAS association colocalises with PTPN1 eQTL signal in thyroid tissue in GTEx v8 dataset (posterior probability for shared causal variant $PP_4=0.87$; Figure 2, Supplementary Data 2), PNPT1 exon expression QTL in monocytes ($PP_4=0.93-0.96$), RP11-554J4.1 eQTL in multiple tissues ($PP_4=0.84-0.92$), and CCDC104 exon expression QTL in fat and blood ($PP_4=0.81-0.91$).”

Colocalisation analysis tests if two independent association signals (such as that from pernicious anemia GWAS and that from an eQTL study) at a locus are consistent with having a shared causal variant. If two traits share a causal variant (they are colocalised), this increases the evidence that they also share a causal mechanism. So if the signal for pernicious anemia colocalises with an eQTL, this could be evidence for this gene’s role in the disease pathogenesis. We use COLOC for colocalisation analysis, which enumerates every possible configuration of causal variants for each of two traits, and calculates the support for that causal model in the form of a Bayes factor can be calculated under an assumption that at most one causal variant per trait exists in the region. The configurations (H_0 - no association; H_1 - association to trait 1 only; H_2 - association to trait 2 only; H_3 - association to both traits, distinct causal variant; H_4 - association to both traits, shared causal variant) (<https://journals.plos.org/plosgenetics/article?id=10.1371/journal.pgen.1008720>). When evaluating the colocalisation of signals, we report PP_4 , which corresponds to H_4 , i.e. likelihood of a shared causal variant in the locus for both traits.

7. What would be the relevance for coloc analysis with thyroid tissue for the chr2 variant associated with B12 deficiency anemia? Would evaluations using blood cell data or gut tissue data be more relevant? Although PNPT1 shows strong coloc posterior probability with expression in the thyroid, equally strong or stronger signals are also seen for CCDC104 in blood cell lines and fat.

*Thank you for this comment! The coloc analysis has now been updated with the latest eQTL catalogue datasets, and for the chromosome 2 association we found the pernicious anemia GWAS association colocalises with PTPN1 eQTL signal in thyroid tissue in GTEx v8 dataset (posterior probability for shared causal variant $PP_4=0.87$; Figure 2, Supplementary Table 2), PNPT1 exon expression QTL in monocytes ($PP_4=0.93-0.96$), and CCDC104 exon expression QTL in fat and blood ($PP_4=0.81-0.91$). On its own this is indeed quite ambiguous and therefore we also looked at functional annotation of the credible set variants and mouse knockout phenotypes for these genes. Credible set analysis showed two variants (rs7586115 and rs13420929, $r=0.7$ with lead signal rs12616502) overlap with PNPT1 transcription start site/flanking region or enhancer marks in several cell types and tissues, including T-cells subtypes (Figure 3). Data from mouse knockouts supports PTPN1 as the most likely candidate causal gene in this locus as *Pnpt1^{tm1a(KOMP)Wtsi/Pnpt1+}* mice exhibit increased mean corpuscular volume (MCV) together with increased mean corpuscular hemoglobin (MCH). Increased red blood cell MCV is a common feature in macrocytic anemias, both megaloblastic (caused by B12 deficiency and pernicious anemia) and nonmegaloblastic (caused by diseases such as myelodysplastic syndrome and hypothyroidism). Furthermore, literature search showed disrupted*

PNPT1 function causes accumulation of mitochondrial RNA in the cytoplasm, which leads to immune activation. In most severe forms, this presents as a group of disorders known as type I interferonopathies, which are commonly characterised by autoinflammation and autoimmunity.

Initially we exhibited the coloc result in thyroid tissue because this region has previously been associated with hypothyroidism.

8. It will be also helpful to show and comment on the results of colocalization analysis for other loci.

All colocalisation results are now shown in Supplementary Data 2, and discussed in more detail in the Results section for loci where colocalisation analysis was the main line of evidence for candidate gene mapping.

9. Are the results at the three HLA alleles fully correlated? It will be helpful to provide the result for the HLA-DR15 haplotype. Will conditioning on the effects of the HLA alleles or HLA-DR15 abolish the association signals within the MHC region?

*Thank you for this suggestion! We conditioned the analysis on the three associated alleles (HLA-DQB1*06:02, HLA-DQA1*01:02, and HLA-DRB1*15:01) and still observed residual signal in the locus (rs4148874, $p=3.4 \times 10^{-6}$). On the other hand, when we conditioned the analysis on rs4148874 (lead signal on chr6 in the Estonian data for which we had information on HLA alleles), the association with specific alleles was not significant. Since we do not have information on HLA alleles for other cohorts, and the lead signal in EstBB is not the same as in the meta-analysis, we have decided to present the observed HLA-allele results, but interpret these with more caution, since we have no supporting data from other cohorts.*

10. The look-up of pernicious anemia with other traits/diseases in the Estonian biobank data is interesting. Were any specific SNPs identified from the study associated with these traits/diseases? Could these associations be evaluated in a Mendelian randomization or a Mediation framework to evaluate potential causal path of pernicious anemia with other diseases?

Indeed, we observed associations with several (autoimmune) conditions and also saw pleiotropy of associated loci. To check whether the pernicious anemia associations were driven by concomitant autoimmune diseases (mostly vitiligo and thyroid problems, which were also highlighted as significant in the associated diagnoses analysis), we did a look-up for vitiligo, hypothyroidism and autoimmune thyroid disease associations in our meta-analysis summary statistics (Supplementary Data 5). Since autoimmune diseases can share pathogenic mechanisms, we focused on loci that according to current knowledge do not regulate autoimmune response. For vitiligo, we chose variants annotated to genes associated with melanocyte biology and for thyroid issues we chose the TSHR (thyroid stimulating hormone receptor) and TPO (thyroid peroxidase) loci and others (Supplementary Data 5). None of these variants reached a nominal significance in our pernicious anemia GWAS meta-analysis, confirming that the observed associations are not mainly driven by concomitant autoimmune disease. We think at the current moment MR or similar analyses are of restricted utility, since a) our analysis included a relatively small number of cases; b) our analysis

included the UKBB dataset, which is also a part of many recent large GWAS studies that could potentially be used for MR analyses, and therefore there is sample overlap that would confound the analyses; c) for many of the observed associations, well-powered GWAS analyses have not been conducted, apart from the analyses conducted on UKBB, which again raises the issue of sample overlap.

11. As for the “Time At First Diagnosis” analysis, all the results are not statistically significant, and the results could be just “chance events”.

We have now removed this analysis and the corresponding plot for clarity.

12. Several GWAS has been performed for B12 deficiency (such as Lin et al. Hum Mol Genet 2012) and B12 serum level (such as Grarup et al PLoS Genetics 2013). Should perform a systematic comparison of the results from the current studies with the ones of these previous studies.

This was an excellent suggestion and we have now conducted a systematic comparison between our results and those from Grarup et al (we chose this study as they analyses European ancestry data). Additionally, we included a study evaluating genetic determinants of parietal cell antibody (PCA) positivity in T1D patients (autoimmune destruction of parietal cells plays an important role in pernicious anemia etiopathogenesis). In short, we find none of the variants influencing B12 levels in the general population were genome-wide significant in our data, but 8/11 showed nominal association p-values, with the allele decreasing serum B12 levels consistently also increasing the risk of pernicious anemia. Similarly, six of the nine T1D risk loci showing significant PCA associations were nominally significant in our meta-analysis (Supplementary table ...), including rs2476601 (PTPN22), which is one of our top associated variants. The associated loci have a central role in immune regulation (incl PTPN22, CTLA4, IFIH1, HLA region and SH2B3). Notably, we did not see an association with the parietal cell autoantibody-associated INS locus, which again suggests that the genome-wide significant loci we report are not confounded by accompanying autoimmune disease (such as T1D where INS plays a central role in the etiopathogenesis).

13. It will be good to include the rsid of lead SNPs in the Manhattan plot on fig1 and the description of ICD codes in Fig3b for easy reference.

Thank you for this comment! The plot has now been updated accordingly.

REVIEWERS' COMMENTS

Reviewer #1 (Remarks to the Author):

The paper now mentions several potential confounding factors. As the authors mention, most of these would serve bias toward the null i.e., decrease the likelihood of seeing a positive result.

Minor points.

1) Y axis labels on Fig 2 "B12 GWAS" confusing as these are PA GWAS data and the paper refers to other GWAS studies in which the genetics of vitamin B12 levels were looked at. More appropriate y-axis labels would add clarity.

2) The authors use data from a mouse knockout to bolster their case for PTPN1 as being the PA gene behind the association signal. They reference the MGI database as claiming that mice carrying a knockout allele of Ptpn1 (genotype Pnpt1tm1a(KOMP)Wtsi/Pnpt1+), "exhibit increased mean corpuscular volume (MCV) together with increased mean corpuscular hemoglobin (MCH)" When one checks the data in MGI, it does list an increased MCV and MCH as being associated with this allele but only in male mice. In order to find the primary data, I went to the KOMP phenotype database. This database does not report any significant hematologic phenotype for the mice carry this same allele. I believe the raw data for this phenotype is also available. While it is possible that I am using the data incorrectly, the authors should attempt to verify their statement to ensure that they are not reaching a conclusion in error.

Reviewer #2 (Remarks to the Author):

The authors have address my points.

Reviewer #3 (Remarks to the Author):

Most of my comments as well as the comments from other reviewers have been addressed. However, I would suggest that authors should soften the claim on the Chr 2, giving the facts that the current result is just below the threshold of genome-wide significant in the meta-analysis of the two datasets, but the surrounding/most likely causal variants did not show any evidence in Finnish dataset. I would argue that the finding remains to be "suggestive" and needs to be further validated by future studies.

Reviewer #1 (Remarks to the Author):

The paper now mentions several potential confounding factors. As the authors mention, most of these would serve bias toward the null i.e., decrease the likelihood of seeing a positive result.

Minor points.

1) Y axis labels on Fig 2 "B12 GWAS" confusing as these are PA GWAS data and the paper refers to other GWAS studies in which the genetics of vitamin B12 levels were looked at. More appropriate y- axis labels would add clarity.

Thank you for noticing this! We have now updated the y-axis labels.

2) The authors use data from a mouse knockout to bolster their case for PTPN1 as being the PA gene behind the association signal. They reference the MGI database as claiming that mice carrying a knockout allele of Ptpn1 (genotype Pnpt1tm1a(KOMP)Wtsi/Pnpt1+), "exhibit increased mean corpuscular volume (MCV) together with increased mean corpuscular hemoglobin (MCH)"

When one checks the data in MGI, it does list an increased MCV and MCH as being associated with this allele but only in male mice. In order to find the primary data, I went to the KOMP phenotype database. This database does not report any significant hematologic phenotype for the mice carry this same allele. I believe the raw data for this phenotype is also available. While it is possible that I am using the data incorrectly, the authors should attempt to verify their statement to ensure that they are not reaching a conclusion in error.

Thank you for pointing this out, we were not aware of the inconsistencies between the two databases. We have now looked up the raw data in International Mouse Phenotyping Consortium (<https://www.mousephenotype.org>) webpage

*(https://www.mousephenotype.org/data/charts?accession=MGI:1918951&allele_accession_id=MGI:4364657&pipeline_stable_id=M-G-P_001&procedure_stable_id=M-G-P_016_001¶meter_stable_id=M-G-P_016_001_005&zygosity=heterozygote&phenotyping_center=WTSI). It seems the data has been presented only for mutant mice so indeed we are not able to report a comparison with wild-type mice. At the same time, from the presented data (which we have now included as Supplementary Data 4 and Supplementary Figure 4) it seems the effect of mutant allele on mean corpuscular volume is more pronounced in male mice, as in the background strain mean corpuscular volume is comparable in male and female mice (Raabe, B. M., Artwohl, J. E., Purcell, J. E., Lovaglio, J. & Fortman, J. D. Effects of weekly blood collection in C57BL/6 mice. *J. Am. Assoc. Lab. Anim. Sci.* **50**, 680–5 (2011)). Inspired by this observation, we conducted a sex-stratified GWAS analysis and meta-analysis for the sentinel signals at each locus in the EstBB and UKBB data, for which we have access to individual level data. These analyses revealed a sexually dimorphic effect for the lead signal on chromosome 2 on the risk of pernicious anemia (OR=2.22(1.63-3.00) in men, OR=1.39(1.15-1.67) in women, GWAMA gender heterogeneity p-value 0.01; Supplementary Table 1). No other lead signal exhibited sexual dimorphism (Supplementary Table 1). We have now included these results in the manuscript as well and clarified the part on mouse phenotypes.*

Reviewer #2 (Remarks to the Author):

The authors have address my points.

We would like to thank the Reviewer for their helpful comments throughout the review process.

Reviewer #3 (Remarks to the Author):

Most of my comments as well as the comments from other reviewers have been addressed. However, I would suggest that authors should soften the claim on the Chr 2, giving the facts that the current result is just below the threshold of genome-wide significant in the meta-analysis of the two datasets, but the surrounding/most likely causal variants did not show any evidence in Finnish dataset. I would argue that the finding remains to be "suggestive" and needs to be further validated by future studies.

*We have now rephrased the part about finding on chromosome 2 and in the concluding paragraph state: "In summary, our analysis of 2,166 cases and 659,516 controls identifies robust risk loci for pernicious anemia in or near candidate genes with a known role in autoimmune conditions (PTPN22, HLA, IL2RA, AIRE) and suggests PNPT1 as a potential causal gene with possible sexually dimorphic effects in the 2p16.1 locus that needs further validation." Additionally we have now found papers that suggest PNPT1 plays a role in both hematopoiesis and immune regulation via TET2 regulation cascade. TET2 is an important factor related to hematopoiesis and immune regulation (Cong, B., Zhang, Q. & Cao, X. The function and regulation of TET2 in innate immunity and inflammation. *Protein and Cell* **12**, (2021)), and it was demonstrated that dysregulation of TET2 involving PNPT1 leads to autoimmune diabetes (Stefan-Lifshitz, M. et al. Epigenetic modulation of β cells by interferon- α via PNPT1/mir-26a/TET2 triggers autoimmune diabetes. *JCI insight* **4**, (2019)). In our opinion, this further supports*

the role on this particular gene in pernicious anemia etiopathogenesis, but we agree with the reviewer that further validation is needed.